# Factors Driving Sustainable Consumption in Azerbaijan: Comparison of Generation X, Generation Y and Generation Z

Mubariz Mammadli 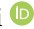

Department of Economics and Business Management, UNEC Business School, Azerbaijan State University of Economics, Baku AZ1001, Azerbaijan; mubariz.mammadli@unec.edu.az

**Abstract:** The importance of sustainable consumption requires understanding and studying the factors that influence consumer preferences. This study contributes to the understanding of inter-generational differences in the factors that drive consumers toward sustainable consumption in Azerbaijan. In this research, 200 sustainable consumers were first interviewed and the factors that pushed them to sustainable consumption were listed. According to the answers received, these factors were ecological concerns, health concerns and subjective norms. Based on these approaches, a survey was conducted among 1380 sustainable consumers in 2022 and analyzed which of these factors had the greater impact among Generations X, Y and Z. The results obtained using ANOVA revealed that ecological and health issues differed across generations, while subjective norms did not vary across generations.

**Keywords:** sustainable consumption; generations; ecological concerns; health concerns; subjective norms

## 1. Introduction

Problems such as the gradual decrease in natural resources important for the world's economic, social and environmental well-being; the disappearance of biological diversity; the increase in environmental pollution; and global warming are among the most important environmental problems in the world. In order to solve these problems, it is essential that societies exhibit sustainable consumer behaviors. However, it should not be forgotten that in the development process of the world, each society has its own dominant values, cultural codes, models of thought and behavior. These values, codes, thought and behavior patterns can also change in certain time patterns within the same society. In other words, it would be useful to examine this issue in the context of generations in order to evaluate sustainable consumer behavior and make various predictions today.

There is always an interaction between generations and the society they live in [1–4]. Each generation is influenced by the social, political and economic values of the society in which they live [5–7]. Differences in the upbringing of generations and the changes taking place in their environment cause significant differences in personal characteristics and life decisions [1,8]. In other words, each generation has its own characteristics, value judgments and attitudes [5]. These differences are also reflected in the consumer behavior of generations.

The first scientific studies on the concept of generation were initiated by the French sociologist Auguste Comte. Comte defined generations as a force acting in the historical process. He stated that social development will take place because of the values that one generation will transfer to the next generation [9]. The division of the population into generations involves the systematization of a group of people born within a certain date range, with similar cultural and social background [10]. This similarity leads to the formation of communities among their perceptions, interests and behaviors. In other words, generations can be defined as people belonging to a common age group, as well as distinct

characteristics such as shared common values, common religious beliefs, common concepts and common lifestyles [3,4,11–13]. The behaviors of these generations differ depending on their adoption of values such as respect for others, interpersonal interaction, level of risk taking, honesty, common sense, responsibility, habits, knowledge, skills, use of technology, motivation, political approach, online socialization, consumption characteristics, health and environmental awareness [14–18].

Today, the generations are divided into six groups according to age characteristics [3,5, 19–22]:

- Silent Generation (born before 1945);
- Baby Boomers (born 1946–1964);
- Generation X (born 1965–1980);
- Generation Y/millennial generation (born 1981–1994);
- Generation Z/post-millennials (born 1995–2010);
- Generation Alpha (born after 2010).

It should be noted that there is no consensus among researchers about the age distribution of generations [23,24]. In particular, the starting year for Gen Z (also known as i-Generation; centennials; zoomers [21]) is sometimes considered to be 1990 [25], while the years 1991 [26], 1994 [27,28] and 1995 [29] are also accepted as the starting year. There are even those who accept the beginning of Generation Z as 1997. For example, Americans born between 1981 and 1996 are included in Generation Y by the Pew Research Center. Those born after 1997 are considered part of Generation Z [21].

In this study, 1995 was chosen as the beginning of Generation Z for Azerbaijan. There are two main reasons for this. The first one is related to the "Contract of the Century" signed on 20 September 1994, after independence. Namely, Azerbaijan's fragile economy started to recover because of this Agreement and foreign investment played a fundamental role in the development of the country's economy. As a result of the Agreement, the "New Oil Strategy" was implemented in Azerbaijan, which became a turning point in the socio-economic conditions of the population [30]. The second reason is the adoption of a new constitution protecting fundamental human rights in 1995. In other words, children born in Azerbaijan in 1995 were born in a new socio-economic era [31].

## 2. Sustainable Consumption and Research Questions of This Study

### 2.1. Sustainable Consumption

Sustainable development is a concept that focuses on the use of resources to the extent necessary to meet the needs of the present, while considering the needs of future generations [28]. Sustainable consumption is one of the most important pillars of sustainable development.

Sustainable consumption responds to people's basic needs and provides them with a better quality of life. It is also the use of goods and services that minimize the use of natural resources, toxic substances, waste and pollutant emissions in a way that does not endanger the needs of future generations [32,33].

Sustainable consumption is expressed by the United Nations as "doing more and better with less" [34]. It is also emphasized that people's sustainable lifestyles reduce harmful effects on the environment and improve well-being. What is expected from consumers in this regard is to reduce waste and be careful about the products they purchase [35]. In other words, consumers are expected to choose sustainable products. In particular, it is recommended not to throw away food, reduce plastic consumption, use reusable bags, etc. It is thought that businesses will have to adopt sustainable solutions because of consumers' conscious purchasing [36].

The phrase sustainable consumption also includes the terms "ethical" [13,37], "green" [38] or "responsible" [39] consumption. In addition, sustainable consumption in a broad sense includes both the purchase of green products and actions to avoid the use, disposal and even overconsumption of products [12,30]. Sustainable consumption helps to ensure better production and an efficient and inclusive supply of food and agriculture. This is because it

encourages agri-food systems to be more resilient and sustainable under changing climatic and environmental conditions [40]. Selecting food products that are clearly eco-labeled and display the logo of good accreditation programs is one of the principles of sustainable, quality nutrition [41]. Sustainable food consumption requires a change in daily eating habits, the purchase of seasonal products [42] and a transition from a predominantly animal diet to a plant-based diet. For example, some studies suggest that reducing meat consumption will positively affect various areas of sustainability, such as health, the environment and biodiversity conservation [43–45]. Sustainable consumer behavior depends on consumers being aware of their long-term impact on the social environment as well as the natural environment [46–48]. In this respect, sustainable consumer behavior involves ensuring that consumers meet certain standards. These standards include the following [49–53]:

- ○ Meeting basic human needs;
- ○ Reducing consumption;
- ○ Realizing responsible and efficient consumption with future generations in mind;
- ○ Placing more emphasis on quality of life than material standards;
- ○ Protecting personal health;
- ○ Making choices in favor of socially and environmentally friendly consumers;
- ○ Minimizing waste and pollution.

According to some researchers, it is difficult for sustainable consumer behavior to spontaneously occur, and additional efforts are required for its development [54,55]. Sustainable consumption is influenced by various factors such as rules in society, the social impact of multilateralism, consumer priority, values, identity building, concern for the environment and protection of personal health [56,57]. At the same time, some studies highlight that younger consumers are more interested in sustainable behavior compared to older ones [58]. It is important to raise awareness and knowledge about various activities to transform society into pro-environmentally sustainable consumers. This need justifies the need for systematic consumer research, including those from Generations X, Y and Z.

Ensuring sustainable consumption and production is one of the 17 Sustainable Development Goals [59]. To achieve these goals, "National Priorities for Socio-Economic Development: Azerbaijan—2030: National Priorities for Socio-Economic Development" was adopted in 2021 and various applications were made [60]. For example, as of 1 January 2021, polyethylene bags with a thickness of up to 15 microns were banned in the country. At the same time, as of 1 July 2021, the sale and distribution of disposable plastic stir sticks, spoons, forks, knives, plates and cups to consumers in commercial, catering and other service facilities were prohibited [61].

### 2.2. Research Questions of This Study

As we mentioned earlier, the factors that drive consumers to sustainably consume, vary. It is seen that the factors affecting sustainable consumer behavior in Azerbaijan have not been examined based on generations. The novelty and originality of this study stem from this deficiency. Therefore, research to address this deficiency is of great importance. The purpose of this study is to identify the factors that contribute to sustainable consumption in Generations X, Y and Z. Since there are always inconsistencies between the factors that affect consumer sustainable consumption [62], identifying the factors influencing sustainable consumer consumption across generations in Azerbaijan will shed light on the implementation of relevant policies.

To prepare the main statements of this study, the following question was asked to consumers who shop in large supermarkets, especially supermarkets such as Bravo, Araz, Bazarstore, Grandmarket, Megastore, Neptun, Bolmart and Rahat market: "Are you a sustainable consumer?". A total of 200 people who identified themselves as sustainable consumers were asked to answer the question "In what ways do you consider yourself a sustainable consumer?". The responses were—consumption of green-labeled products, use of products with environmental symbols, use of recycled products, using less plastic and avoiding overconsumption.

These consumers were then asked the question "Why do you prefer sustainable consumption?". The distribution of the answers obtained as a result of the first phase of this study, which was carried out in the form of individual interviews in January 2022, was as follows (Table 1):

**Table 1.** Ratio of factors driving sustainable consumption.

| Statement | Percent |
| --- | --- |
| To protect my health. | 35% |
| For ecological cleaning. | 33% |
| To protect nature. | 26% |
| This is the "fashion" of today. | 21% |
| Since it is GMO-free, it does not bloat my body and I feel happy. | 16% |
| For peace of mind. | 13% |
| To control my weight/My dietitian recommended it. | 12% |
| In order not to be condemned by loved ones. | 10% |

Based on the data obtained, the factors that stimulate sustainable consumption were systematized under three headings: ecological concern, health concern and subjective norms. In accordance with these headings, the main questions addressed within the scope of this research were determined as follows:

1. Does ecological concern, which is a triggering factor in the realization of sustainable consumption, differ between generations X, Y and Z?
2. Does health concern, which is a triggering factor in the realization of sustainable consumption, differ between X, Y and Z generations?
3. Do subjective norms, which are a triggering factor in the realization of sustainable consumption, differ between generations X, Y and Z?

## 3. Literature Review and Hypothesis Development

In this section, the theoretical background and the main hypotheses of this research are discussed.

### 3.1. Environmental Concern Approach to Sustainable Consumption

One of the main reasons consumers turn to sustainable consumption is their interest in the environment and the importance they attach to environmental protection policies [63–65]. Environmental concern is the approach of individuals to the dangers that arise in the world and their concerns about the effects of these dangers on nature and future generations [66,67].

The literature argued that individuals with a high level of environmental concern generally show environmentalist behaviors in their purchasing activities. These people, who take protective actions toward nature, are knowledgeable about the harm of products to the environment and are sensitive to environmental pollution [68].

For example, analyzing the approaches of 530 participants from Portugal, Fontes et al. found that concern for nature, environmental attitudes and behavior are important indicators of environmentally friendly purchasing decisions [69].

Kement [70] investigated the effect between environmental concern and the desire to use green hotels through an empirical study involving 391 green hotel visitors and concluded that concern for nature and accepted moral duty positively affect the desire to use green hotels.

Borusiak et al. [71] studied the impact of Polish consumers' concern for the environment on their actual behavior and intentions to minimize water consumption from disposable plastic bottles. The results of this study confirmed the indirect impact of concern

for the environment on both intentions and behavior regarding the consumption of bottled water. Environmental concern was found to be positively associated with attitudes toward reducing bottled water consumption.

Chen, Wu and Jiang [72] investigated the impact of environmental concern on ecological purchasing behavior. The results showed that consumers' growing concern for the environment has a positive impact on their eco-purchasing behavior. In other words, ecological responsibility has a direct impact on ecological purchasing behavior. Another study found that environmental concerns and the idea of recycling have a significant influence on obtaining children's clothing produced using ecological cotton [73].

In terms of generations, some studies show that young people are more sensitive to ecological problems. For example, Nikoli'c et al. [74] highlighted that in Bosnia and Herzegovina and Serbia, Generation Z is more aware of recycling and is concerned about the future of the planet. Another study was carried out by Prakash and Pathak [75] on the data of young Indian consumers. The results of the research revealed that people's desire to obtain eco-friendly packaging is greatly affected by environmental concerns. Moreover, research by Bastounis et al. [76] found that younger participants were more sensitive to eco-labels than older participants. In a circular economy study, it was found that in Romania, Generation Z's attitude toward the circular economy is statistically weaker than Generation X's, both in terms of waste sorting and recycling in general [77]. Joshi and Rahman [78] investigated the psychological factors predicting sustainable purchasing behavior among 425 educated young consumers. The researchers' results showed that environmental responsibility was the key psychological determinant of consumers' sustainable purchasing decisions. On the other hand, Tan and Lau [79] examined the effect of the general environmental attitude of undergraduate students in Malaysia on green purchasing behavior. The results obtained by these researchers show that environmental attitude does not make a significant contribution to green purchasing behavior.

Thus, there is increasing evidence in the literature that ecological concern promotes sustainable consumption [80,81]. To analyze whether this approach is valid for Azerbaijani consumers and whether it differs between generations, "green purchasing attitudes" used in the study of Joshi and Rahman [78] were evaluated as the main measure of ecological concern. For this purpose, three statements were used in the questionnaire. These expressions are given below:

○ Reducing pollution and protecting the environment are key factors in realizing sustainable consumption.
○ Reducing the waste of natural resources is one of the main factors in realizing sustainable consumption.
○ Protection of the natural environment is one of the basic factors in realizing sustainable consumption.

Thus, we can present the first hypothesis of this study as follows:

**Hypothesis 1 (H1).** *Ensuring sustainable consumption due to ecological concerns varies between generations.*

**H 1.1.** *Realization of sustainable consumption to reduce pollution and protect the environment differs between generations.*

**H 1.2.** *Realization of sustainable consumption to reduce the waste of natural resources differs between generations.*

**H 1.3.** *Realization of sustainable consumption for the protection of the natural environment differs between generations.*

*3.2. Health Care Approach to Sustainable Consumption*

According to the Food and Agriculture Organization, green products include products that provide food and nutrition and contribute to the healthy living of all generations [82].

Green products also include diets that minimize adverse environmental impacts. Therefore, the consumption of green products has a positive impact on the environment by promoting food recycling and reuse, mitigating climate change and reducing waste [83,84]. Regardless of the environmental benefits of organic foods, some researchers emphasize that these foods are consumed for health reasons [85–87]. On the other hand, it is also mentioned that to achieve food security in society, healthy nutrition must be ensured. In addition, improving eating habits will allow for obtaining the necessary amount of dietary energy and ensuring an active life. Consuming healthy products will be effective in preventing micronutrient deficiencies, obesity, cardiovascular diseases, diabetes and certain types of cancer [88].

In the research conducted by Nguyen et al. [89], who investigated the attitudes of consumers toward green products, it is revealed that the approach of consumers changes according to their health awareness and environmental concerns. The researchers also observe that consumers care more about their health than environmental issues when making a purchase decision. In Makatouni's research [90], it is underlined that parents give more priority to green products in terms of children's health and safety.

As a result of the research conducted by Verain et al. in Niderland in 2014, it was found that explaining the benefits of both health and sustainability positively affected people's thoughts about sustainable behaviors [91]. The research conducted by Barska et al. shows that Generation Y consumers in Poland pay attention to product quality, price, freshness and taste, as well as the opinions of dietitians and nutritionists, when choosing innovative food products [92].

Consumer motivation to buy organic products was also studied by Zonoli and Naspetti [93] in Italy. As a result of the study, even though environmentally friendly products are considered expensive and difficult to obtain, the majority of consumers who participated in the study rated the product positively. In addition, consumers associated organic products with health. In another study conducted by Coderoni et al. [94], it was found that consumers pay attention to the nutritional contributions of organic fruits and therefore prefer these foods as healthy products.

The study conducted by Gazdecki et al. shows that some consumers attach great importance to the health benefits obtained as a result of sustainable consumption [95]. At the same time, meat consumption among sustainability-conscious consumers decreases when the benefits of health and sustainability concepts are taken into account [91].

One of the largest health concerns today is weight control. Overweight or obesity is caused by a variety of factors—genetic, social, economic, environmental, political, physiological, urbanization and technological [96]. In addition, diets are one of the important factors affecting the level of overweight or obesity [86]. One of the main approaches to sustainable consumption is to change people's unhealthy consumption habits that cause overweight and obesity [97].

According to the standard accepted by the World Health Organization, a person with a body mass index (BMI) of 25.0 to 30.0 is considered "overweight". Obesity reflects an average BMI $\geq$30.0, i.e., the presence of abnormal or excessive fat accumulation that can impair health [98]. In 2016, 39% of adults worldwide were overweight or obese. At the same time, the proportion of children and adolescents aged 5–19 who are overweight or obese increased from 4% in 1975 to about 18% in 2016 [98]. According to the World Obesity Federation 2023 Atlas, it is predicted that 51 percent of the world's population, that is, more than 4 billion, will face obesity or overweight disease in the next 12 years [99].

It is possible to prevent obesity by following a healthy diet throughout your life [100]. Synthetic fertilizers, pesticides, growth hormones and pesticides are not used in the production of organic products [101]. Eating organic reduces the amount of chemicals in the body [102]. Therefore, consumers believe that organically labeled products are healthier than conventional products. Since the consumption of organic products is more beneficial in terms of health, it will have a positive impact on protecting human health [103–106].

On the other hand, according to some studies, being overweight reduces happiness, which negatively affects mental health [107–109]. This shows that being overweight affects the mental health as well as the physical health of the person. One of the important issues that people today find difficult to protect is mental health. In recent years, there has been a significant increase in mental disorders, including depression, gloom, delusion and sleep disorders. Factors influencing the occurrence of these disorders in humans include diet and biologically active components [110].

According to a study by Venhoeven, Boulderdijk and Steg [111], the commitment to environmentally friendly behavior affects how people see themselves and form their feelings. Moreover, consumers who do not buy environmentally friendly products experience psychological discomfort and may feel guilty [112]. Mazar and Zhong [113] emphasize that people behave more altruistically after consuming environmentally friendly products compared to traditional goods. However, consumers tend to be less altruistic and more prone to fraud and theft after consuming environmentally friendly goods compared to traditional goods.

Thus, considering that consumers prefer sustainable consumption for health reasons, it has been examined whether this situation is valid between X, Y and Z generations in Azerbaijan. In addition, given the importance consumers attach to health, attempts have been made to determine which health reason is important in the realization of this behavior among those who make sustainable consumption. For this purpose, three statements were used in the questionnaire. These expressions are given below:

○　Physical health is one of the key factors in realizing sustainable consumption.
○　Weight control is one of the main factors in achieving sustainable consumption.
○　Mental health is one of the main factors in realizing sustainable consumption.

The research hypothesis is prepared as follows.

**Hypothesis 2 (H2).** *Ensuring sustainable consumption due to health concerns varies between generations.*

**H 2.1.** *Realization of sustainable consumption due to physical health changes from generation to generation.*

**H 2.2.** *Realization of sustainable consumption due to weight control changes from generation to generation.*

**H 2.3.** *Realization of sustainable consumption due to mental health changes from generation to generation.*

*3.3. Subjective Norms Approach to Sustainable Consumption*

People's attitudes are defined as their positive or negative evaluation of their own actions in relation to a particular behavior. Various studies have shown a positive impact between consumers' attitudes and their willingness to purchase green [114,115].

Human behavior is influenced by subjective norms [116,117]. The subjective norm is that a person behaves in a certain way because of social pressure [118].

Subjective norms are thought to be defined by some existing normative beliefs about the anticipations of various persons, especially family members, relatives, friends and co-workers [63,119].

A study by Yadav and Pathak [120] attempted to understand the intention of young consumers to buy organic products in India as a developing country. The result showed that the intention of young consumers to buy green products is primarily influenced by environmental concerns, but also by subjective norms, environmental knowledge and other factors. A similar result was obtained in the study conducted by Robichaud and Yu. Namely, subjective norms or peer influence was found to be effective on the sustainable consumption of Generation Z [13].

Mahasuweerachai et al. conducted one of the studies investigating the relationship between perceived social value and feelings of guilt and pride. Namely, according to the findings of this study, social value has a great impact on feelings of guilt and pride. Generation Z's sense of guilt and pride affects their eating habits in restaurants [121].

Prakash and Pathak [75] analyzed the data of young Indian consumers and found that personal norms and attitudes significantly influence their intention to purchase environmentally friendly packaging. King and Dennis [118] showed in their research that consumers' attitudes and subjective norms are among the variables that affect the shopping abandonment process.

Although some studies have found that the relationship between consumers' attitudes, subjective norms and perceived behavioral control and sustainable consumption is significant [64,122], another group found that subjective norms do not have a positive correlation to the purchase of green products [123].

A study carried out by Rausch and Kopplin [67] tested whether there was a positive relationship between subjective norms and the desire to purchase environmentally friendly clothing. The analysis was based on three statements—"*My friends expect me to buy sustainable clothes*", "*My family expects me to buy sustainable clothes*" and "*People who are important to me expect me to buy sustainable clothes*". As a result of the analysis, the positive impact of subjective norms on the purchasing intention of sustainable clothing has not been proven.

A survey conducted by Sousa et al. [119] showed that among university students in Portugal, subjective norms did not have an affirmative impact on students' intentions to buy environmental goods.

Kumar et al. [124] mentioned that there is no significant connection between subjective norms and willingness to buy green products in a collectivist culture in India.

According to this study, the direction of the subjective norm as a moderator in the formation of environmental knowledge and attitudes is not supported.

Thus, the approaches of Yadav and Pathak [120] to subjective norms were used in the preparation of the third hypothesis of the study to assess consumer attitudes toward sustainable consumption. This hypothesis is as follows:

**Hypothesis 3 (H3).** *Ensuring sustainable consumption due to subjective norms varies between generations.*

**H 3.1.** *Realization of sustainable consumption varies from generation to generation under the influence of "people whose opinions the consumer considers".*

**H 3.2.** *Realization of sustainable consumption varies from generation to generation under the influence of "people who are important to the consumer".*

**H 3.3.** *Realization of sustainable consumption varies from generation to generation under the influence of "people who like the consumer".*

## 4. Material and Methods

The "Sample size calculator" of the SurveyMonkey online survey portal was used to determine the minimum number of people to participate in the survey [125]. Data from the State Statistical Committee of the Republic of Azerbaijan reveal that as of 1 January 2022, the population of Azerbaijan was 10,156.4 thousand people [126]. According to SurveyMonkey's calculation, a minimum of 385 people should take part in this survey, with a confidence level of 95% in accordance with this population.

Simple random sampling method was used to save time in accessing these data. Simple random sampling equalizes the probability that each member of the population is part of the sample. The basic logic in this method is to eliminate the bias in the selection process and to obtain representative patterns [127]. In general, for the simple random method to be applied correctly, the sample size should be over a few hundred [128].

The study population consists of representatives of Generations X, Y and Z living in different regions of Azerbaijan. The research survey was conducted twice. The first stage of the research consisted of interviews, as mentioned in Section 2.2 of this study, and the second stage consisted of a questionnaire survey based on the results of the first stage. The second stage was conducted between January and November 2022.

In the second stage of this study, people consuming in large supermarkets were first asked "Are you a sustainable/environmentally friendly consumer?". People who considered themselves sustainable consumers were asked to fill out the questionnaire survey.

In the first part of the questionnaire survey, consumers' demographic characteristics such as age, gender, profession, education level, region of residence, and religious belief were asked. Since the focus of this article is Generations X, Y and Z, the analysis was made only on the age criterion. Other demographic indicators will be analyzed in another article. Moreover, in order not to spoil the data set, we did not attempt to equalize the number of people included in the X, Y and Z generations.

Table 2 presents the data obtained according to the sample characteristics of the variables.

**Table 2.** Sample characteristics.

| Generation | Number | Percent % |
|:---:|:---:|:---:|
| X Generation | 427 | 30.9 |
| Y Generation | 499 | 36.2 |
| Z Generation | 454 | 32.9 |
| Total | 1380 | 100 |

According to the results presented in Table 2, 30.9% of the participants belong to Generation X, 36.2% to Generation Y and 32.9% to Generation Z.

The data in Table 3 reflect the characteristics of the participants regarding why they consider themselves as sustainable consumers.

**Table 3.** Participants' behavior as sustainable consumers.

| Statement | Percent |
|:---|:---:|
| Consumption of green labeled products | 62% |
| Use of products with eco friendly symbols | 59% |
| Use of recycled products | 31% |
| Less use of plastics | 28% |
| Avoid overconsumption | 23% |
| Other | 24% |

Table 3 reveals that 62% of participants consider themselves to be sustainable consumers as they consume green-labeled products, and 59% prefer products with eco-friendly symbols. Additionally, 31% of respondents use recycled products, while only 28% prefer to use less plastics and 23% avoid overconsumption.

In the third part of the survey, consumers choose which of the displayed factors (ecological concerns, health concerns and subjective norms) pushed them toward sustainable consumption.

Table 4 shows that 30.4% of the participants sustainably consume due to ecological concerns, 44.8% due to health concerns and 24.8% due to subjective norms. It should also be noted here that 14 participants were not included in the analysis, because they selected "other" option when answering this question. Moreover, the data of 175 people were not assessed, because they were not suitable for the age axis of this study.

**Table 4.** Participants' behavior as sustainable consumers.

| Statement | Number | Percent % |
|---|---|---|
| Ecological concern | 420 | 30.4% |
| Health concern | 618 | 44.8% |
| Subjective norms | 342 | 24.8% |
| Total | 1380 | 100 |

IBM SPSS Statistics Version 26 program was used to evaluate the collected data.

## 5. Findings

In this section, the results of the ANOVA of the factors affecting sustainable consumption according to generations are given and attempts were made to prove the hypotheses of this research.

Before performing ANOVA, it is necessary to determine whether the data have a normal dispersion. Whether there is a normal distribution among quantitative variables can be evaluated based on several tests. Skewness and kurtosis coefficients were used to examine whether the variables had a normal dispersion.

The skewness value reflects the level of symmetry present in a variable's dispersion. The kurtosis value is a scale that indicates whether the difference between the dispersion is excessively high [129]. If it takes values between −1 and +1, it shows that the skewness and kurtosis coefficients have a normal distribution.

In Table 5, the values obtained through the skewness and kurtosis analysis are given. The obtained results show that these values are between −1 and +1 and an ANOVA test can be carried out.

**Table 5.** Skewness and kurtosis coefficients of the variables.

| Descriptives | | Statistic | Std. Error |
|---|---|---|---|
| Ecological concern approach | Skewness | 0.018 | 0.456 |
| | Kurtosis | −0.173 | 0.520 |
| Health approach | Skewness | −0.558 | 0.321 |
| | Kurtosis | −0.795 | 0.584 |
| Subjective norms approach | Skewness | 0.622 | 0.398 |
| | Kurtosis | −0.543 | 0.464 |

### 5.1. Data Analysis

Before starting the ANOVA analysis, a reliability analysis among variables was measured using the Cronbach alpha method. A Cronbach's alpha value greater than 0.70 shows that this scale is reliable. If the value of Cronbach alpha is below 0.70, then the reliability of these scales is considered low or unreliable.

The Cronbach alpha value calculated for the variables is given in Table 6, and it revealed that the reliability values for each variable were high.

**Table 6.** Reliability statistics.

| | Cronbach's Alpha | Cronbach's Alpha Based on Standardized Items | N of Items |
|---|---|---|---|
| Ecological concern approach | 0.819 | 0.825 | 3 |
| Health approach | 0.874 | 0.874 | 3 |
| Subjective norms approach | 0.793 | 0.793 | 3 |

*5.2. Verification of Hypothesis 1*

In order to prove the first hypothesis, the ecological concern approach to sustainable consumption was investigated by generation.

Table 7 shows the ANOVA results related to the ecological concern approach dimension.

**Table 7.** ANOVA results regarding ecological approach.

| | | Sum of Squares | df | Mean Square | F | Sig. |
|---|---|---|---|---|---|---|
| Reducing pollution and protecting the environment is one of the key factors in the realization of sustainable consumption. | Between Groups | 1736.564 | 2 | 868.282 | 617.517 | 0.000 |
| | Within Groups | 1937.586 | 1378 | 1.406 | | |
| | Total | 3674.151 | 1380 | | | |
| Reducing the waste of natural resources is one of the main factors in the realization of sustainable consumption. | Between Groups | 2691.217 | 2 | 1345.609 | 1103.047 | 0.001 |
| | Within Groups | 1681.024 | 1378 | 1.220 | | |
| | Total | 4372.242 | 1380 | | | |
| Protection of the natural environment is one of the main factors in the realization of sustainable consumption. | Between Groups | 1213.254 | 2 | 606.627 | 581.488 | 0.001 |
| | Within Groups | 1437.574 | 1378 | 1.043 | | |
| | Total | 2650.828 | 1380 | | | |

When examining Table 7, it can be seen that "Sig." values for all specified components are less than 0.05. This result shows that the H1 hypothesis is accepted. In other words, it is confirmed with 95% certainty that ecological concern as one of the main drivers of sustainable consumption varies between generations.

The one-way ANOVA test does not show how the ecological concern approach changes from generation to generation. Therefore, a Bonferroni post hoc test is carried out.

Based on the results of the Bonferroni post hoc test, it can be said that there is a difference between the approaches of Generation Z and Generation X (see Appendix A). That is, Generation Z viewed the options presented in terms of sustainable consumption more positively. In other words, it has been found that while Generation Z consumes sustainably to reduce pollution and protect the environment, reduce the waste of natural resources and protect the natural environment, Generation X does not take the same approach to these factors. Generation Y has also been very inclined to reduce the waste of natural resources in the realization of sustainable consumption, unlike the X generation.

*5.3. Verification of Hypothesis 2*

To prove Hypothesis 2, three statements were considered from the perspective of generations. The results of the analysis of variance regarding whether health concerns differ according to generation are shown in Table 8. The data show that the "Sig." value is less than 0.05 for all expressions. These data reveal that the preference for sustainable consumption due to health concerns differs on the basis of generation.

**Table 8.** ANOVA results regarding health approach.

| | | Sum of Squares | df | Mean Square | F | Sig. |
|---|---|---|---|---|---|---|
| Physical health is one of the main factors in realizing sustainable consumption. | Between Groups | 1541.942 | 2 | 770.971 | 3891.476 | 0.000 |
| | Within Groups | 273.006 | 1378 | 0.198 | | |
| | Total | 1814.949 | 1380 | | | |
| Weight control is one of the main factors in realizing sustainable consumption. | Between Groups | 524.938 | 2 | 262.469 | 584.585 | 0.000 |
| | Within Groups | 618.700 | 1378 | 0.449 | | |
| | Total | 1143.638 | 1380 | | | |
| Mental health is one of the main factors in realizing sustainable consumption. | Between Groups | 291.878 | 2 | 145.939 | 670.631 | 0.001 |
| | Within Groups | 299.873 | 1378 | 0.218 | | |
| | Total | 591.751 | 1380 | | | |

Results from the Bonferroni post hoc test show that the main difference is between Generation Z and Generations X and Y (see Appendix B). Generation X and Y's approaches to implementing sustainable consumption for physical health have been found to be positive compared to Generation Z. On the other hand, the implementation of sustainable consumption through weight control is positively perceived by Generation Z and Y compared to Generation X. In other words, Generation Z and Y realize sustainable consumption for weight control; however, a similar approach was not seen in Generation X. Mental health, as a factor that guides sustainable consumption, has also had a more positive effect among the members of Generation Z, unlike Generation X.

*5.4. Verification of Hypothesis 3*

The values shown in Table 9 indicate that there is no difference between subjective norm approaches. This means that there was no significant difference between the averages of consumers' attitudes toward preferring sustainable consumption according to subjective norms. As a result of ANOVA analysis, H2 was rejected.

**Table 9.** ANOVA results regarding subjective norms approach.

| | | Sum of Squares | df | Mean Square | F | Sig. |
|---|---|---|---|---|---|---|
| Most of the people whose opinion I consider prefer that I buy green products. | Between Groups | 13.206 | 2 | 6.603 | 3.280 | 0.038 |
| | Within Groups | 2774.430 | 1378 | 2.013 | | |
| | Total | 2787.636 | 1380 | | | |
| Most of the people important to me would like me to buy green products. | Between Groups | 40.730 | 2 | 20.365 | 8.702 | 0.284 |
| | Within Groups | 3224.912 | 1378 | 2.340 | | |
| | Total | 3265.642 | 1380 | | | |
| Most of the people who like me think I should buy green products. | Between Groups | 302.020 | 2 | 151.010 | 116.847 | 0.676 |
| | Within Groups | 1780.897 | 1378 | 1.292 | | |
| | Total | 2082.917 | 1380 | | | |

**6. Discussion**

Today, the widespread use of sustainable consumption is one of the crucial phenomena to be achieved in society. In this regard, people who prefer sustainable consumption are the focus of this study and their behavior is analyzed. However, the main purpose of this research is to investigate the reasons that push people to consume this way and to find out whether there are differences on the basis of generations.

During the analysis, it was found that people in Azerbaijan generally gravitate toward sustainable consumption based on three approaches: ecological concern, health concern and subjective norms. The findings of this research are as follows: (1) the ecological concerns differ according to the generations, (2) the health concerns differ according to the generations and (3) subjective norms do not differ according to the generations.

It has been determined that the Z generation is more sensitive to environmental problems than the X generation. In other words, the Z generation tends to consume in a sustainable way to reduce pollution and protect the environment compared to the X generation. This result is similar to the results obtained by Nikoli'c et al. [74], Prakash and Pathak [75], and Bastounis et al. [76] and reveals that Gen Z is more sensitive to ecological concerns.

In addition, another difference between generations in the realization of sustainable consumption has emerged in the approach of reducing the waste of natural resources. In other words, Generation Y showed a more positive tendency toward this statement compared to Generation X.

The data obtained revealed that the Z generation is more sensitive to environmental problems, environmental protection and reducing pollution and sustainably consumes due to these concerns. Based on these data, it can be said that it would be useful to expand the policy of informing non-sustainably consuming Gen-Zers about environmental issues and emerging problems. Environmental education plays an important role in understanding the magnitude of environmental problems and experiencing positive changes in people's environmental behavior.

Likewise, considering that the attitude of the Y generation toward the waste of natural resources is more evident and they consume them in a sustainable way to prevent this situation, we can say that raising the awareness of all individuals of this generation on the prevention of waste will increase their attitudes toward sustainable consumption in a positive way. As we know, the media influence what readers think about, what they perceive as important and how they deal with the problems with the content they create. In other words, preventing the waste of natural resources and increasing awareness of environmental issues will especially encourage Y and Z generations to consume sustainable products.

Another striking finding in this study is that some consumers turn to sustainable consumption because of the importance they attach to their health. This result is similar to those of other studies such as Gazdecki [95], Verain et al. [91] and Zonoli and Naspetti [93]. However, what sets our study apart from other studies is the discovery of which health issues differ across generations in terms of sustainable consumption.

So, in terms of health concerns, Generations X and Y were found to have more positive attitudes toward preferring sustainable consumption for physical health than Generation Z. This result shows that if the connection between sustainable consumption and physical health is explained to all X and Y generations, these generation members will tend toward sustainable consumption.

On the other hand, the realization of sustainable consumption for weight control is perceived more positively by the Z and Y generations compared to the X generation. In other words, the Z and Y generations realize more sustainable consumption for weight control compared to the X generation. Using this information by the companies producing sustainable products and the informative promotions for the Z and Y generations will be the focus of attention of individuals who want to lose weight, as well as those who want to eat healthily and keep their weight under control.

The healthy lifestyle that individuals take part in includes all the behaviors they believe and practice for the purpose of remaining healthy. These behaviors include many different factors such as exercise, nutrition, stress management and spiritual satisfaction, which also affect mental health. When the sustainable consumption behaviors of consumers in Azerbaijan are examined in terms of protecting their mental health, it is observed that there is a significant difference between Generation X and Generation Z. In other words,

the members of Generation Z prefer sustainable consumption more than other generations for the protection of mental health. Considering that the Z generation consumes more sustainable products due to their mental health, it can be assumed that explaining the benefits obtained during the production and consumption of these products to consumers will increase sustainable consumption even more.

On the other hand, research shows that people's high food literacy can help improve ecological eating behavior and eliminate health problems. For example, the results of a study conducted by Lee et al. on 395 university students in South Korea showed that the components of students' food literacy improved ecological eating behavior. Thus, strengthening consumers' food literacy is important for expanding sustainable consumption [130].

Our study revealed that subjective norms are among the factors that push sustainable consumption. This result is similar to studies conducted by Rausch and Kopplin [67], Prakash and Pathak [75], King and Dennis [118], and Yadav and Pathak [120]. Although there are subjective norms among the reasons respondents prefer sustainable consumption, there is no significant difference between generations in terms of the priority of this factor. Even though there is no difference between generations, the fact that consumers in Azerbaijan sustainably consume under the influence of people who care about and consider their opinions, as well as people who like them, does not change. In other words, the increase in the proportion of people who sustainably consume in the country will lead to an increase in the influence of subjective norms on people.

It is very difficult to change people's food preferences, choices and eating habits. Additionally, there is a significant gap between consumers' positive attitudes and sustainable consumption [131].

## 7. Conclusions

The results of this study revealed differences between the factors determining whether consumers prefer sustainable consumption across generations. These differences can be used by public policy and sustainable product manufacturers within the scope of various applications and sustainable consumption can be encouraged with different projects.

The needs and desires of Generation X, Y and Z consumers are affected by many factors, including biological, psychological, social, economic and cultural factors. To better understand the needs of consumers, it will be useful for manufacturers and various businesses, especially advertising creators, to know consumer behavior and the reasons for this behavior. Considering the factors that consumers care about, especially when introducing new products to the market or making changes to existing products, will help increase sustainable consumption. It would be particularly effective to utilize these and similar research results and for market players to consider factors such as age appropriateness, special needs and expectations when developing products and services to be launched on the market.

On the other hand, more effort is currently needed to increase consumers' sustainability awareness, because consumer decisions are some of the most important factors in shaping the market environment. To implement such policies, considering the characteristics of consumers and the values they care about by policy makers will serve to make sustainable consumption more widespread among people. Especially, according to the results obtained in this article, it is possible to produce and promote products suitable for the profiles of consumer segments and to motivate segments that are more sensitive.

*Limitations and Future Research*

There are some limitations regarding the scope of this study. Namely, it could have been analyzed in more depth by considering the high rate of health reasons that push consumers toward sustainable consumption. However, this area was not expanded, in order to narrow the scope of this study. Considering that the majority of consumers consume sustainably for health reasons, more details can be investigated in the future. Useful policies can be followed in line with the results obtained. Additionally, more research

is needed on Generation X, Y and Z consumers to develop a sustainable consumption model in the future.

**Funding:** This research received no external funding.

**Institutional Review Board Statement:** The study was conducted in accordance with the Declaration of Helsinki and approved by the Institutional Review Board (or Ethics Committee) of Economic Think (protocol code 301518 and date of approval 25 February 2022).

**Informed Consent Statement:** Informed consent was obtained from all subjects involved in the study.

**Data Availability Statement:** The data presented in this study are available on request from the corresponding author.

**Conflicts of Interest:** The author declares no conflict of interest.

## Appendix A

**Table A1.** Bonferroni post hoc test results regarding ecological approach.

| Multiple Comparisons | | | | | | | |
|---|---|---|---|---|---|---|---|
| **Bonferroni** | | | | | | | |
| Dependent Variable | (I) Generation | (J) Generation | Mean Difference (I − J) | Std. Error | Sig. | 99% Confidence Interval | |
| | | | | | | Lower Bound | Upper Bound |
| Reducing pollution and protecting the environment is one of the key factors in the realization of sustainable consumption | Generation X | Generation Y | 1.83891 | 0.07408 | 0.014 | 1.6614 | 2.0165 |
| | | Generation Z | −1.04487 * | 0.07102 | 0.000 | −1.2151 | −0.8746 |
| | Generation Y | Generation X | −1.83891 | 0.07408 | 0.014 | −2.0165 | −1.6614 |
| | | Generation Z | −2.79894 | 0.08314 | 0.260 | −2.9982 | −2.5997 |
| | Generation Z | Generation X | 1.04487 * | 0.07102 | 0.000 | 0.8746 | 1.2151 |
| | | Generation Y | 2.79894 | 0.08314 | 0.260 | 2.5997 | 2.9982 |
| Reducing the waste of natural resources is one of the main factors in the realization of sustainable consumption | Generation X | Generation Y | −1.80987 * | 0.06900 | 0.001 | −1.9753 | −1.6445 |
| | | Generation Z | −1.80946 * | 0.07547 | 0.000 | −1.9904 | −1.6286 |
| | Generation Y | Generation X | 1.80987 * | 0.06900 | 0.001 | 1.6445 | 1.9753 |
| | | Generation Z | −3.61933 | 0.07744 | 0.588 | −3.8049 | −3.4337 |
| | Generation Z | Generation X | 1.80946 * | 0.07547 | 0.000 | 1.6286 | 1.9904 |
| | | Generation Y | 3.61933 | 0.07744 | 0.588 | 3.4337 | 3.8049 |
| Protection of the natural environment is one of the main factors in the realization of sustainable consumption | Generation X | Generation Y | −1.21864 | 0.06381 | 0.206 | −1.3716 | −1.0657 |
| | | Generation Z | −2.35269 * | 0.06979 | 0.000 | −2.5200 | −2.1854 |
| | Generation Y | Generation X | 1.21864 | 0.06381 | 0.206 | 1.0657 | 1.3716 |
| | | Generation Z | −1.13405 | 0.07161 | 0.092 | −1.3057 | −0.9624 |
| | Generation Z | Generation X | 2.35269 * | 0.06979 | 0.000 | 2.1854 | 2.5200 |
| | | Generation Y | 1.13405 | 0.07161 | 0.092 | 0.9624 | 1.3057 |

*. The mean difference is significant at the 0.01 level.

# Appendix B

**Table A2.** Bonferroni post hoc test results regarding health approach.

| Multiple Comparisons | | | | | | | |
|---|---|---|---|---|---|---|---|
| **Bonferroni** | | | | | | | |
| | | | | | | **99% Confidence Interval** | |
| **Dependent Variable** | **(I) Generation** | **(J) Generation** | **Mean Difference (I − J)** | **Std. Error** | **Sig.** | **Lower Bound** | **Upper Bound** |
| Physical health is one of the main factors in realizing sustainable consumption | Generation X | Generation Y | −2.17002 | 0.02781 | 0.190 | −2.2367 | −2.1034 |
| | | Generation Z | 2.14729 * | 0.03041 | 0.000 | 2.0744 | 2.2202 |
| | Generation Y | Generation X | 2.17002 | 0.02781 | 0.190 | 2.1034 | 2.2367 |
| | | Generation Z | 1.06033 * | 0.02914 | 0.002 | 0.9905 | 1.1302 |
| | Generation Z | Generation X | −2.14729 * | 0.03041 | 0.000 | −2.2202 | −2.0744 |
| | | Generation Y | −1.06033 * | 0.02914 | 0.002 | −1.1302 | −0.9905 |
| Weight control is one of the main factors in realizing sustainable consumption | Generation X | Generation Y | −1.38574 * | 0.04186 | 0.001 | −1.4861 | −1.2854 |
| | | Generation Z | −1.29483 * | 0.04579 | 0.000 | −1.4046 | −1.1851 |
| | Generation Y | Generation X | 1.38574 * | 0.04186 | 0.001 | 1.2854 | 1.4861 |
| | | Generation Z | −1.09091 | 0.04698 | 0.073 | −1.2035 | −0.9783 |
| | Generation Z | Generation X | 1.29483 * | 0.04579 | 0.000 | 1.1851 | 1.4046 |
| | | Generation Y | 1.09091 | 0.04698 | 0.073 | 0.9783 | 1.2035 |
| Mental health is one of the main factors in realizing sustainable consumption | Generation X | Generation Y | 0.02273 | 0.03121 | 1.000 | −0.0521 | 0.0975 |
| | | Generation Z | −1.61715 * | 0.03188 | 0.001 | −1.6936 | −1.5407 |
| | Generation Y | Generation X | −0.02273 | 0.03121 | 1.000 | −0.0975 | 0.0521 |
| | | Generation Z | −0.44318 | 0.03271 | 0.057 | −0.5216 | −0.3648 |
| | Generation Z | Generation X | 1.61715 * | 0.03188 | 0.001 | 1.5407 | 1.6936 |
| | | Generation Y | 0.44318 | 0.03271 | 0.057 | 0.3648 | 0.5216 |

*. The mean difference is significant at the 0.01 level.

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
