# Peer review of "Factors Driving Sustainable Consumption in Azerbaijan: Comparison of Generation X, Generation Y and Generation Z"

_sustainability, doi:10.3390/su152015159_

Round 1
Reviewer 1 Report
Subject: The manuscript „ FACTORS DRIVING SUSTAINABLE CONSUMPTION: COMPARISON GENERATION X, GENERATION Y AND GENERATION Z”
Well-presented paper which confirms previous studies in this space. My concern on the paper is that I didn’t find what measure constructs was used in survey, is it previously established and validated scales? Such consideration should be done with the research methodology and method ought to be made clear.
Author Response
Dear reviewers,
Thank you for your comments.
Your comments have had a great impact on improving the quality of the article. I thank you all for the privilege.
The comments you noted have been corrected in the article.

Reviewer 2 Report
The manuscript titled "Factors driving sustainable consumption: Comparison of Generation X, Generation Y, and Generation Z" presents an insightful exploration into the different factors influencing sustainable consumption across diverse generations, focusing on the contextual setting of Azerbaijan. The paper is generally well-structured, providing substantial theoretical grounding and a clear articulation of the concepts of generations and sustainable consumption. However, some areas require revision and clarification for a more cohesive and thorough presentation.
1. Similarity Index: The similarity index of the manuscript is notably high at 44%. It is crucial that the author(s) address this issue promptly and effectively, revising the manuscript to reduce similarity and ensure the originality and integrity of the academic contribution.
2. Definition and Classification of Generations: The manuscript delineates clear definitions and classifications of the generations under study. However, it is imperative to address the disparities and lack of a universal consensus on the age distribution of generations and justify the classifications employed in this study adequately.
3. Definition of Sustainable Consumption: The manuscript offers a detailed exploration of sustainable consumption, outlining the standards and considerations inherent to it. However, these definitions need to be clear, precise, well-referenced, and integrated into the study's overall framework, providing a solid foundation for the research.
4. Practical Implications: The practical implications of sustainable consumer behavior should be discussed in more depth. The manuscript should address how the findings can inform policy, practice, and consumer choices and contribute to the promotion of sustainability in the contextual setting of Azerbaijan and potentially beyond.
Author Response

(The authors gave the same response as above.)

Author Response
Dear reviewers,
Thank you for your comments.
Your comments have had a great impact on improving the quality of the article. I thank you all for the privilege.
The comments you noted have been corrected in the article.
I made corrections to the literature. However, there may be some errors that are overlooked due to lack of time. Generally, editorial team help resolve this issue and make necessary corrections after the manuscript is accepted.

Reviewer 4 Report
The topic taken up by the authors is very interesting. However, the article contains some errors and shortcomings that should be taken into account before possible publication.
1. The title should include information that the research concerns only one selected area, i.e. Azerbaijan.
2. The abstract is too general and does not contain the required elements. Please expand it accordingly, according to the journal's requirements.
3. In my opinion, the layout of the article and the substantive content contained in some sections are incorrect - the Introduction should be placed first, then the Literature Review - and only on the basis of the literature review should the gaps be pointed out, which will consequently allow the formulation of research problems and hypotheses - and these elements should be included in the Materials and Methods section. However, in this article it is scattered - first in Section 2 there are Research Questions of the Study (and partly the results of preliminary research?), then there is Section 3 - titled Literature Review and Hypothesis Development and only Section 4 Material and Methods.
4. There is also a lack of clearly indicated novelty and originality of the research conducted, as well as the purpose of the article - again, this information is scattered throughout the text and must be searched for.
5. The authors write in lines 322-323 that "Simple random sampling method was used to save time in accessing these data.", but there is no information on how the research objects included in the research sample were randomly selected. What was the sampling frame?
6. The Material and Method section lacks specific information about the research sample or research tool used. There is also no indication or description of the research methods and techniques used - was it some direct survey method? Survey technique? Was the tool a research questionnaire? If so, it should be described in detail.
7. The Findings section describes some results, but how were they obtained? What was the raw data?
8. The Discussion and Conclusions section contains only a discussion of our own results, there is no comparison with the research of other authors from other countries, especially since the authors write that "As mentioned earlier, the factors that push consumers towards sustainable consumption differ in every society ".
It seems to me that the entire text from this section could as well be included in the previous section - commenting on the results obtained.
9. The article contains a sufficient number of references, although it should be noted that some of them are not very current.
10. I am not qualified to assess the quality of English in this paper, but it seems to me that care should be taken to ensure linguistic, grammatical and stylistic correctness throughout the article.
Author Response
Dear reviewers,
Thank you for your comments.
Your comments have had a great impact on improving the quality of the article. I thank you all for the privilege.
The comments you noted have been corrected in the article.
3. The main reason why the Literature review section is given after the Research question section in the article should be due to the fact that the Literature review section is shaped in line with these questions. For example, since the research question did not cover the religious affiliation of consumers, this topic was not included in the Literature review section.

Round 2
Reviewer 3 Report
Dear Authors,
thank you for considering my suggestions and I appreciate your response. The presented version of the article is more correct. After reviewing the content of the new Sustainability- 2628094 ‘Factors driving sustainable consumption: Comparison of Generation X, Generation Y and Generation Z, would like to conclude that the issue addressed is in line with the theme of the journal. The problem presented in the article is important and interesting, the research in this area is needed.
The structure of the publication is appropriate, the goals and hypothesis are clearly defined. The methodology used is good, correctly described and supported by literature. The conclusions of the analysis presented are correct and in line with the stated objectives. More literature has been added. Conclusion and Future Areas of Research: more text has been added.
I was very glad to learn about your achievements.
Good luck with your next research!
Author Response
Thank you for your attention!
Reviewer 4 Report
The authors responded and reacted to some of the comments, making corrections to the text, but not all of them. It is also difficult for me to find everything in the current version of the manuscript because instead of the Authors' responses file, I see the file with the manuscript.
1. I still think that the layout of the article and the substantive content of some sections are incorrect - the Introduction should be placed first, then the Literature Review - and only on the basis of the literature review should the gaps be indicated, which will consequently allow for the formulation of research problems and hypotheses - these elements should be found in the Materials and Methods section.
2. There is still no clear indication of the novelty and originality of the research conducted.
3. There is still no information on how the research objects included in the research sample were selected. Is the sample representative?
4. I still stand by my comments regarding the Materials and Methods and Findings sections.
Author Response
Dear reviewer, Thank you for your patience.
Replies to your comments were mistakenly uploaded to another reviewers section. I added it again.
